# Single-crystal 2D covalent organic frameworks for high-capacity methane storage

Baoqiu Yu [1,2], Felipe L. Oliveira [3,4], Wenliang Li[5], Qingmei Xu[2], Xu Ding[1], Shangwei Yuan[1], Yucheng Jin [2], Hua Liu[1], Hailong Wang [2]✉, Xin Xiao[1]✉, Jingping Zhang[5], Guillaume Maurin [3,4], Banglin Chen [6,7]✉ & Jianzhuang Jiang [1,2]✉

2D covalent organic frameworks (COFs) usually possess a polycrystalline nature as well as lower porosity and surface area than 3D counterparts, restraining their exploration over gas storage applications. Herein, a substituent strategy has been proposed and employed to generate three robust single-crystal 2D COFs isomers with atom-resolution structures determined by 3D electron diffraction. Among three isomers, a precise engineering of their interlayer distance affords the highest Brunauer−Emmett−Teller surface area of ~2100 m² g⁻¹ and the largest pore volume of 1.40 cm³ g⁻¹ for the desolvated GZU-1. This COF shows the highest total volumetric methane uptake of 240 cm³ (STP) cm⁻³ at 273 K and 100 bar among 2D COFs, even comparable with those for excellent 3D MOFs. This work not only delivers unique insight into the design of 2D single-crystal COFs by interlayer stacking regulation, but also promotes the application of highly porous 2D COFs in gas storage.

Natural gas (NG) with about 95% methane (CH₄) component is a clean and widely available energy source with relatively low carbon emission and economic advantage[1–5]. The low energy density of CH₄, 0.04 MJ L⁻¹, retards the application of NG in onboard transportation fuel[2–6]. Among potential materials and technologies for high methane storage, porous sorbents have shown bright promise, particularly under moderate pressure and ambient temperature, attributed to their extraordinarily high porosities and variable pore structures[1,7–10]. Accordingly, a number of three-dimensional metal-organic frameworks (3D MOFs) have been reported to exhibit high performance for methane storage[3–7]. MOFs are generally robust and can sustain thermal/vacuum activation to keep their high porosities[3,11]. Meanwhile, covalent organic frameworks (COFs) can

also exhibit high porosities and stability[2,12–14]. However, the covalent bonds in COFs are much more directional and less flexible than the coordination bonds in MOFs, meaning that single-crystal COFs are much more difficult to synthesize than MOFs[15–31] and remain scarcely explored for methane storage to date[2,10,12,32].

Different from their MOFs counterparts dominated by the three-dimensional (3D) extended frameworks, most of the reported COFs are of two-dimensional (2D) layered structures[1,33–36]. This seems to retard the exploration of their methane storage application due to the still lack of reports on high methane storage performance for 2D MOFs associated with the low porosities and instability of the latter materials. Fortunately, the late construction of highly mesoporous 2D COFs with pore volume as high 2.7 cm³ g⁻¹ motivates us to explore 2D COFs for

[1]Guizhou Key Laboratory of Macrocyclic and Supramolecular Chemistry, School of Chemistry and Chemical Engineering, Guizhou University, Guiyang, China. [2]Beijing Key Laboratory for Science and Application of Functional Molecular and Crystalline Materials, Department of Chemistry and Chemical Engineering, School of Chemistry and Biological Engineering, University of Science and Technology Beijing, Beijing, China. [3]ICGM, Université de Montpellier, CNRS, ENSCM, Montpellier, France. [4]Institute Universitaire de France (IUF), Paris, France. [5]Faculty of Chemistry, Northeast Normal University, Changchun, China. [6]Fujian Provincial Key Laboratory of Polymer Materials College of Chemistry & Materials Science, Fujian Normal University, Fuzhou, China. [7]Key Laboratory of the Ministry of Education for Advanced Catalysis Materials, College of Chemistry and Materials Science, Zhejiang Normal University, Jinhua, China. ✉e-mail: hlwang@ustb.edu.cn; xxiao@gzu.edu.cn; banglin.chen@fjnu.edu.cn; jianzhuang@ustb.edu.cn

their methane storage functionality[35]. Additionally, the significant progress achieved on the single-crystal growth of COFs and single-crystal structure determination technologies over the past few years also motivates us to reveal the fundamental science of 2D/3D COFs in a more comprehensive manner[16–19]. However, it is still challenging to prepare 2D single-crystal COFs due to the uncontrollable nucleation and complicated crystallization processes. Only three examples, including Py-1P[37–39], COF-400[36], and oDMPA-Py-COF[40] with atom-resolution 2D single-crystal structures have been directly obtained from monomer polymerization thus far. In-depth exploration of the growth and structure of 2D COFs single crystals is very difficult, further restraining the structure-originated property and functionality investigation. Previous investigations have revealed that changing the planarity of building blocks through substituents could lead to the preparation of high-quality 3D COFs single crystals[16,41].

Herein, along the substituent strategy, we report three highly porous, robust 2D single-crystal COF isomers for their methane storage (Fig. 1), reaching total volumetric methane storage up to 240 cm³ (STP) cm⁻³ at 273 K and 100 bar, and in particular gravimetric methane delivery working capacity of 213 mg g⁻¹ (5 to 65 bar at 298 K). Detailed crystal structure and molecular modeling studies show that multiple weak intermolecular interactions, such as C−H···π interactions, can collaboratively enforce the COF framework stability and thus maximize their porosities by interlayer packing regulation. This work paves the way towards a systematic development of highly porous 2D COFs for gas storage in the future.

## Results and Discussion

### Design and synthesis of single-crystal GZU-1, GZU-2, and GZU-3

In the present case, the methyl or methoxy substituents were introduced onto the positions between the two neighboring phenyl moieties of 1,3,5-tri(phenyl)benzene to afford 1,3,5-trimethoxy-2,4,6-tris(4-aminophenyl)benzene (TOAB), 1,3,5-trimethyl-2,4,6-tris(4-formylphenyl)benzene (TTFB), 1,3,5-trimethyl-2,4,6-tris(4-aminophenyl) benzene (TTAB), and 1,3,5-trimethoxy-2,4,6-tris(4-formylphenyl)benzene (TOFB) (Supplementary Figs. 1–6). Subsequently, [3 + 3] imine polymerization reaction at 298 K in different solvents with the

assistance of aniline and acetic acid afforded three novel 2D single-crystal COFs, GZU-1, GZU-2, and GZU-3 (Fig. 1). GZU-1 was made from TOAB and TTFB in a mixed solution of mesitylene (Mes) and 1,4-dioxane (Diox) for 15 days with a yield of 71%. In contrast, isomeric GZU-2 and GZU-3 were obtained by TTAB and TOFB in tri-chloromethane (TCM) and mixed Mes/1,4-dioxane, respectively, with the yield of 52 and 63%, respectively. Comparison in FT-IR data of these single-crystal COFs and building blocks indicates the disappearance of N−H stretching bonds (3339 and 3420 cm⁻¹ for TOAB and 3338 and 3410 cm⁻¹ for TTAB) and C = O stretching bonds (1702 cm⁻¹ for TTFB and 1698 cm⁻¹ for TOFB). Moreover, the presence of the C=N stretching bonds (1625 cm⁻¹ for GZU-1, 1628 cm⁻¹ for GZU-2, and 1629 cm⁻¹ for GZU-3) suggests the successful imine condensation (Supplementary Figs. 7–9). This is further confirmed by the observation of the resonance peak due to the carbon from the C=N bonds at 160 ppm for GZU-1 and 157 ppm for GZU-2 and GZU-3 in their solid-state ¹³C cross-polarization magic-angle-spinning (CP/MAS) NMR spectra (Supplementary Figs. 10–12). Scanning electron microscopy (SEM) images reveal a rod-shaped morphology of GZU-1 with a length size up to 10 μm and hexagonal prism morphologies of GZU-2 and GZU-3 with a length size up to 5 μm (Fig. 2a–c and Supplementary Figs. 13–15). The more than 10 intense, sharp diffraction peaks for these three COFs highlight their high-crystallinity nature (Fig. 2d–f). In particular, GZU-2 and GZU-3, composed of the same building units, display obviously different PXRD patterns, indicating their isomerism nature (Fig. 2e, f and Supplementary Figs. 16–19).

### Structural determination of GZU-1, GZU-2, and GZU-3

Single-crystal structural analyses of the three COFs isomers were carried out by 3D electron diffraction (ED) at 77 K and resolutions of 0.8 ~ 1.0 Å, which show that they have essentially isomeric imine-linked honeycomb layered architectures with the **hcb-b** topology (Supplementary Figs. 20–22 and Supplementary Table 1–3). They possess similar porous frameworks with the 1D channels (16 ~ 17 Å) along the crystallographic c-axis (Fig. 1). GZU-1, GZU-2, and GZU-3 crystallize in distinct space groups, i.e. $P3_2$, $P$-1, $P6_5$, respectively. There are different numbers of independent imine-bonded 1,3,5-tris(phenyl)benzene-

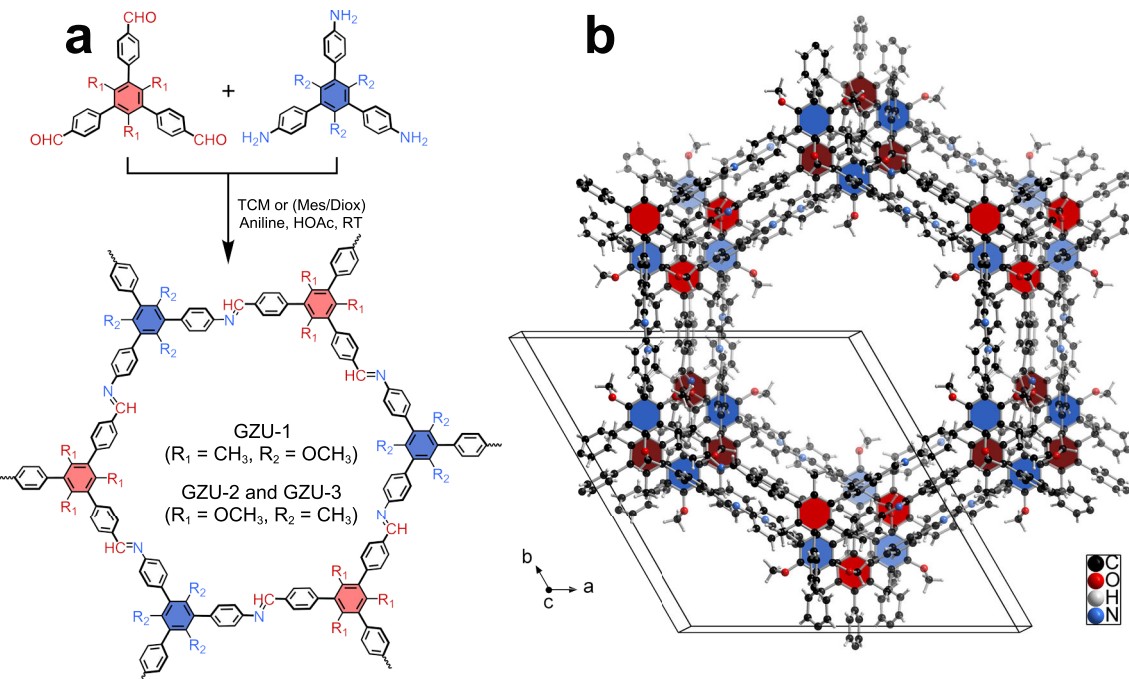

**Fig. 1 | Formation of single-crystal 2D COFs. a** Schematic assembly of single-crystal GZU-1, GZU-2 and GZU-3 2D COFs. **b** A unit cell of GZU-1 containing ABCDEF 6-fold alternating stacking (AA$l_6$) structure with multiple C-H···π interactions between the substituents and frameworks to stabilize its permanent porosity.

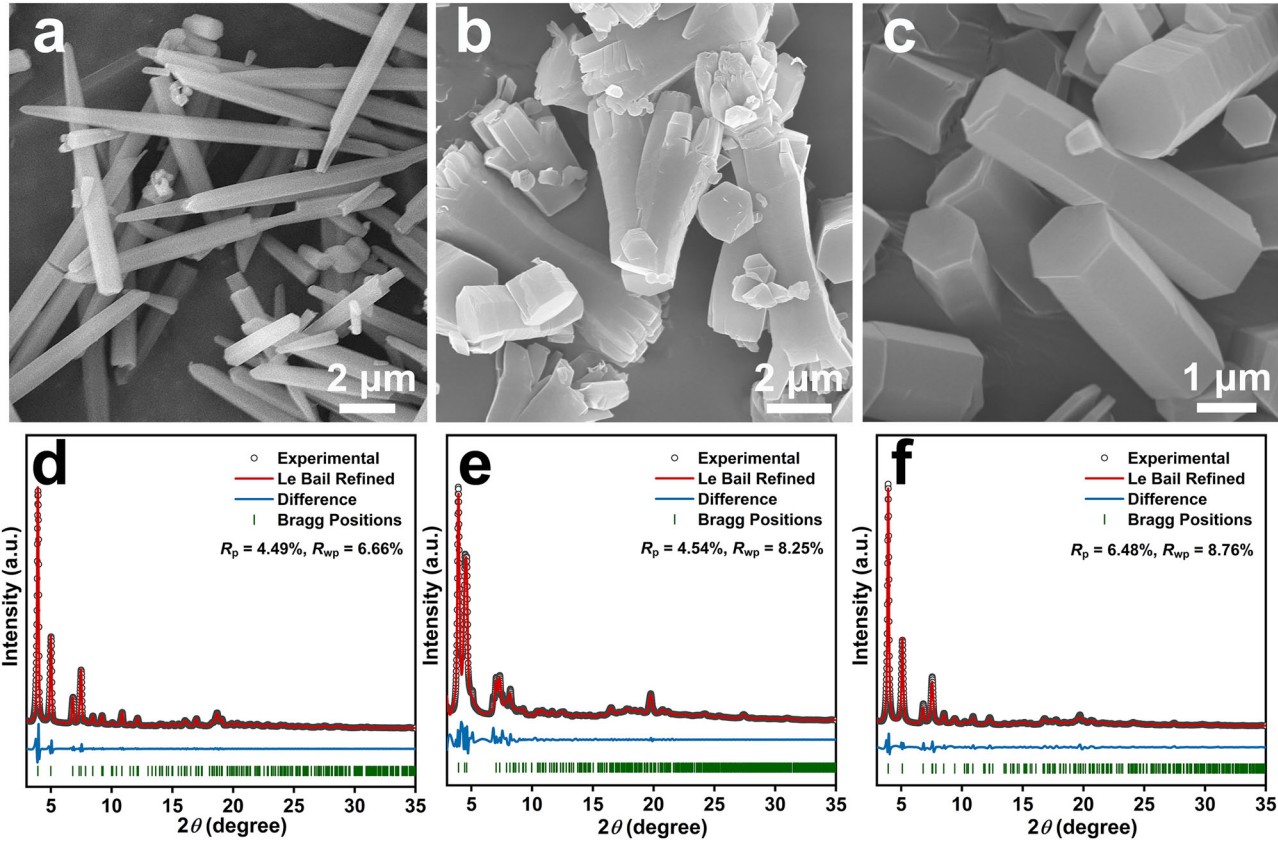

**Fig. 2 | Morphological and structural characterizations of single-crystal 2D COFs. a–c** The SEM images of single-crystal GZU-1, GZU-2, and GZU-3, respectively. **d–f** The Le Bail fitting (red lines) of experimental PXRD patterns (black circles) for activated GZU-1, GZU-2, and GZU-3, respectively.

based building blocks in corresponding asymmetric units of the three COFs.

For GZU-1 derived from TOAB and TTFB, its unit cell parameters are $a = b = 25.600(4)$ Å, $c = 27.550(6)$ Å, $\alpha = \beta = 90°$, $\gamma = 120°$, and $V = 15636(6)$ Å$^3$ (Fig. 3a and Supplementary Table 1). In the asymmetric unit of GZU-1, there are two crystallographically independent TTFB and two TOAB moieties linked through imine bonds (Fig. 3a and Supplementary Fig. 23). For the TTFB units, the dihedral angles between the central benzene ring and the three peripheral benzene rings are 77.94 ~ 82.19°, respectively, disrupting the planarity of this module. This is also true for TOAB units with the dihedral angles of 48.78 ~ 56.33° between central and three peripheral benzene rings (Supplementary Fig. 23). The TOAB and TTFB building blocks are interconnected into a 2D covalent honeycomb-like layered structure (Fig. 3b and Supplementary Figs. 24, 25). Moreover, the alternating arrangement of TTAB and TOFB is observed in the neighboring layers. In this compound, there are multiple weak C–H⋯π interactions between adjacent layers with short H⋯C distances of 2.71 ~ 2.85 Å (Supplementary Fig. 26). Different from the previous 2D COFs with normal AA-, AB-, and ABC-packing structures, GZU-1 shows the unknown ABCDEF 6-fold alternating stacking (AA$l_6$) (Figs. 1, c–e and Supplementary Fig. 27). The neighboring layers crossing the centers of TTFB and TOAB moieties are 4.58 and 4.60 Å (Fig. 3d, e and Supplementary Fig. 27). Density Functional Theory (DFT) calculations explored the relative energy of the different stacking patterns of GZU-1. The results demonstrate that the AA$l_6$ form is the most energetically stable one, being approximately 62, 73, and 122–141 kJ mol$^{-1}$ per sheet more stable than the slip AA$l$, the eclipsed AA, and the staggered AB patterns, respectively, as detailed in Supplementary Fig. 28 and Supplementary Table 4. All DFT-optimized geometries and details of the calculations are provided in the Supplementary Information.

For GZU-2 and GZU-3, these two COF isomers are prepared from the same building blocks of TTAB and TOFB, possessing different imine-orientation from that of GZU-1 (Supplementary Figs. 29, 30). The slight difference in the molecular configuration of 1,3,5-tris(phenyl) benzene-based building blocks with the same substituents is displayed for these three COFs (Supplementary Fig. 31). Similar to the dihedral angles between central and three peripheral benzene rings in TTFB for GZU-1, these values for TTAB unit are in a range of 78.11 ~ 87.45° for GZU-2 and 82.49 ~ 89.19° for GZU-3 (Supplementary Figs. 29 and 30). In contrast, corresponding dihedral angles of TOFB vary in 47.18 ~ 57.52° for GZU-2 and 53.34 ~ 60.42° for GZU-3. However, the numbers and strength of weak C–H⋯π interactions among three neighboring layers in GZU-2 and GZU-3 are different from those of GZU-1. In contrast to GZU-1, there are fewer C–H⋯π weak interactions between adjacent layers in GZU-2 (2.82-2.93 Å) and GZU-3 (2.58 ~ 3.03 Å) (Supplementary Figs. 32 and 33). The distances of neighboring layers crossing the centers of TTAB and TOFB units are 4.37/4.47 Å and 4.35/4.44 Å (Supplementary Figs. 34 and 35). It is worth noting that the gradual decrease of interlayered distance from GZU-1, GZU-2 to GZU-3 slightly changes their pore volume from of 1.41, 1.40, to 1.34 cm$^3$ g$^{-1}$ derived from the single-crystal structures. Furthermore, GZU-2 and GZU-3 exhibit distinctly different packing forms, namely a unidirectional ABCD 4-fold tilted AA stacking (AA$t_4$) structure and AA$l_6$ architecture, respectively (Supplementary Figs. 28, 34, 35). DFT calculations show that these stacking patterns are the most stable among all possible arrangements, with GZU-2 being slightly more stable than GZU-3 by 5 kJ mol$^{-1}$ per sheet. Notably, the preferential packing in GZU-2 is more stable by 66, 74, and 102–135 kJ mol$^{-1}$ per sheet than the AA$l$ slip, AA eclipsed, and the staggered AB patterns, respectively, as shown in Supplementary Fig. 28 and Table 4 (see Supplementary Information for more details of the distinct

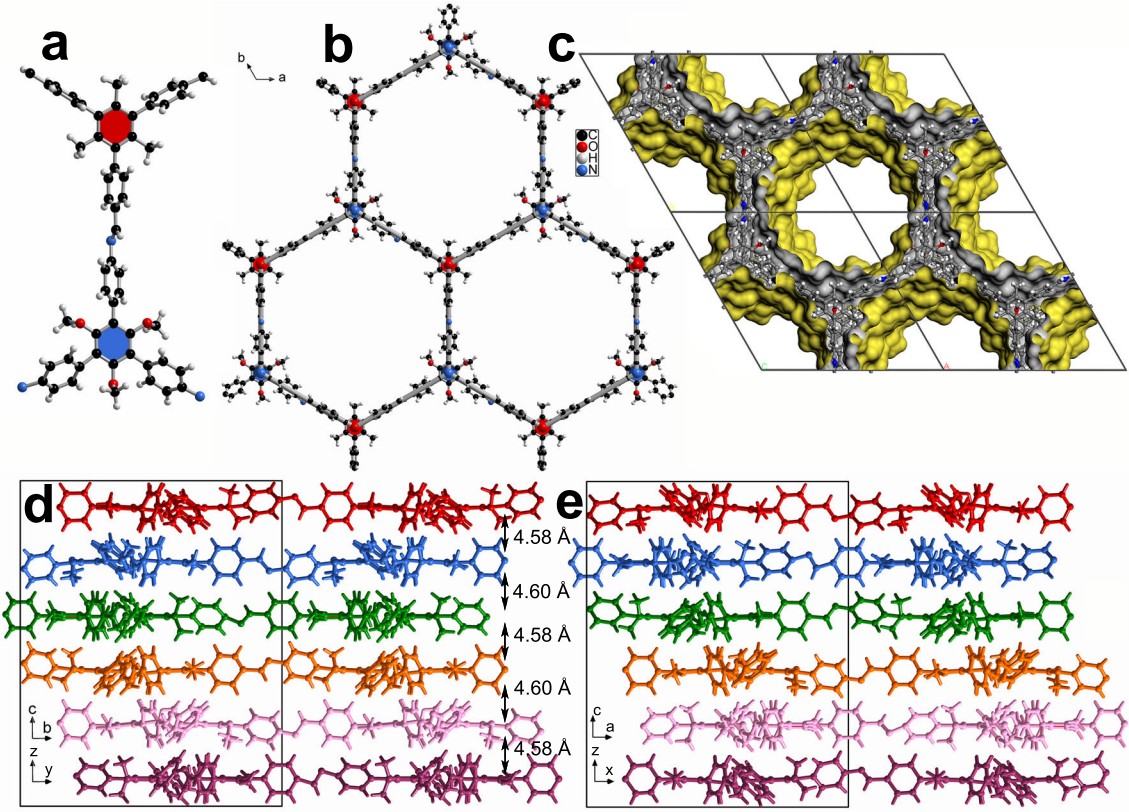

**Fig. 3 | Crystal structure of GZU-1. a** Imine-bonded TOAB and TTFB units in single-crystal GZU-1. **b** A 2D honeycomb layer (the thick grey bond represents **hcb-b** topology). **c** Packing structure of GZU-1 along *c* axis showing the pore surfaces of 1D channels indicated by yellow/gray (inner/outer) curved planes. **d** and **e** The side view of the packing structure along *a* and *b* axis, respectively.

geometries). Interestingly, this energetic stabilization is very similar to that discussed above for GZU-1.

The experimental PXRD patterns of these three COFs align with those derived from the single-crystal structures (Supplementary Figs. 36–38). Following the dynamic vacuum activation, the three activated COFs (GZU-1a, GZU-2a, GZU-3a) exhibit analogous PXRD patterns with parent materials, suggesting a rather rigid behavior of this family of COFs upon desolvation (Supplementary Figs. 36–38). Additionally, the phase purity of all activated GZU-1, GZU-2, and GZU-3 is also demonstrated using whole profile pattern matching by the Le Bail refinement, giving the ultimate $R_p$ and $R_{wp}$ values of 4.49 and 6.66% for GZU-1a, 4.54 and 8.25% for GZU-2a, and 6.48 and 8.76% for GZU-3a, respectively (Fig. 2d–f).

To further elucidate the underlying reasons for the synthesiz-ability and high crystallinity of these single-crystal COFs, DFT calculations were further conducted to assess the relative stability of the structures compared to other similar COFs reported in the literature, such as DISTAP-1[42], TPB-DMTP[24], COF-701[43], TTI-COF[44], and HHTP-DPB[45]. The data presented in Fig. 4 show that GZU-2 exhibits the most favorable formation energy, at −172 kJ mol⁻¹ per sheet, followed by GZU-3 and GZU-1 with −167 and −165 kJ mol⁻¹ per sheet, respectively. Interestingly, these values are substantially more favorable (i.e., more negative) than those presented by the other COFs, which show values of −113, −106, −103, −71, and −59 kJ mol⁻¹ per sheet for TPB-DMTP, DISTAP-1, COF-701, TTI-COF, and HHTP-DPB, respectively. This more favorable formation energy and consequently predicted greater sta-bility of the resulting crystal structures can directly relate to the higher incidence of C−H⋯π interactions between the sheets of the GZU COFs. These interactions arise from the relative displacement presented by the special stacking pattern of these structures. These C−H,...,π interactions combined, which generally exhibit greater stability than the π⋯π eclipsed interactions, confer to the GZU-1, GZU-2, and GZU-3 a greater overall crystal stability[46]. This is ultimately associated with the high-quality single crystals obtained for these materials, a task notably difficult for 2D COFs[47,48]. To further explore the nature of the interac-tion between the COF layers, we conducted a non-covalent interaction (NCI) analysis based on the DFT electronic density. The result shown in Supplementary Fig. 39 for the GZU-1 indicates that there are two main types of interactions between the adjacent COF layers. The first one is the CH₃,...,π interaction between the OMe groups and the aromatic ring from the upper (or lower) layer, with the NCI surface highlighted in green in Supplementary Fig. 39a. The second interaction is the T-shaped benzene-benzene interaction, with the NCI surface high-lighted in yellow. Both interactions that can only occur due to the relative shift between the layers are individually stronger than the simple π⋯π benzene interaction and act as conformational lockers on the crystal structure, playing a decisive role in dictating the relative orientation of the building blocks during crystallization. This indicates the representative structural motif nature of the combination of CH₃,...,π and T−shaped benzene interactions within this family of materials.

**Permanent porosity of activated single-crystal 2D COFs**

The permanent porosities of the activated COFs (GZU-1a, GZU-2a, and GZU-3a) were revealed using N₂ adsorption and desorption measure-ments at 77 K (Fig. 5a and Supplementary Figs. 40 and 41). The N₂ adsorption isotherms of these three COFs display a pronounced two-step uptake behavior. A sharp increase in adsorption isotherms is observed at very low relative pressures (P/P₀ <0.01), followed by a more gradual uptake in the range of P/P₀ ≈ 0.01−0.05. Upon further

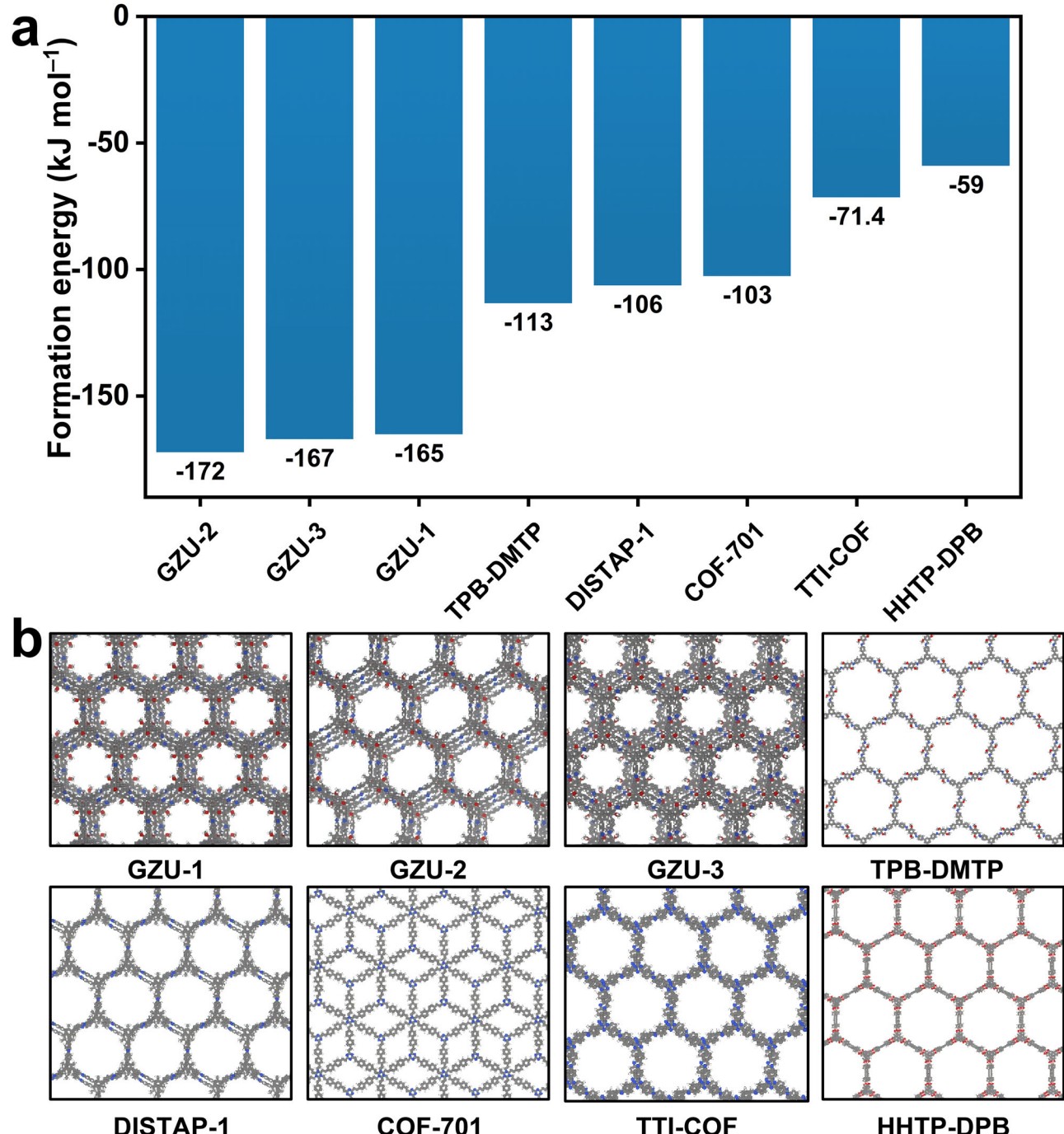

**Fig. 4 | Formation energy comparison. a** Formation energies in kJ mol$^{-1}$ per COF sheet and **b** atomic representation of the structures explored.

increasing the relative pressure, the other sharp increase of adsorption appears in the range of $P/P_0 \approx 0.05$–$0.1$, achieving a saturated adsorption. For GZU-1a with the largest interlayer distance among the three COFs, it also displays the largest $N_2$ uptake of 917 cm$^3$ g$^{-1}$ at 1.0 bar with the highest BET area of 2100 m$^2$ g$^{-1}$, in line with the calculated value from the crystal structure of 2414 m$^2$ g$^{-1}$ (Supplementary Fig. 42).

In contrast, GZU-2a with the second interlayered distance adsorbs smaller $N_2$ volume of 836 cm$^3$ g$^{-1}$ at 1.0 bar with a slightly lower BET area of 1965 m$^2$ g$^{-1}$, close to its simulated value of 2396 m$^2$ g$^{-1}$ (Supplementary Figs. 43 and 44). Its packing isomer, GZU-3a, with the smallest average interlayered distance, accommodates the least $N_2$ amount of 820 cm$^3$ g$^{-1}$ at 1.0 bar with the smallest BET surface area of 1897 m$^2$ g$^{-1}$, consistent with the simulated value of 2326 m$^2$ g$^{-1}$. Interestingly, there exists an overall linear relationship between the average interlayered distance and experimental BET surface area for the three COFs, indicating the capability of the packing mode and imine orientation of 2D COFs isomers in finely engineering the interlayered distance and in turn the pore surface area (Supplementary Fig. 45). It is worth noting that the present single-crystal COFs possess much higher BET areas (1897-2100 m$^2$ g$^{-1}$) than their unsubstituted COF counterpart TAPB-TFPB (-224 m$^2$ g$^{-1}$)[49], further confirming the advantage of substituent strategy in stabilizing the permanent porosity. Following the increased interlayer distance from GZU-3a, GZU-2a, and GZU-1a, their experimental pore volume at $P/P_0$ at 1.0 bar also gets increased in a gradual manner from 1.23, 1.28, to 1.40 cm$^3$ g$^{-1}$, overall consistent with

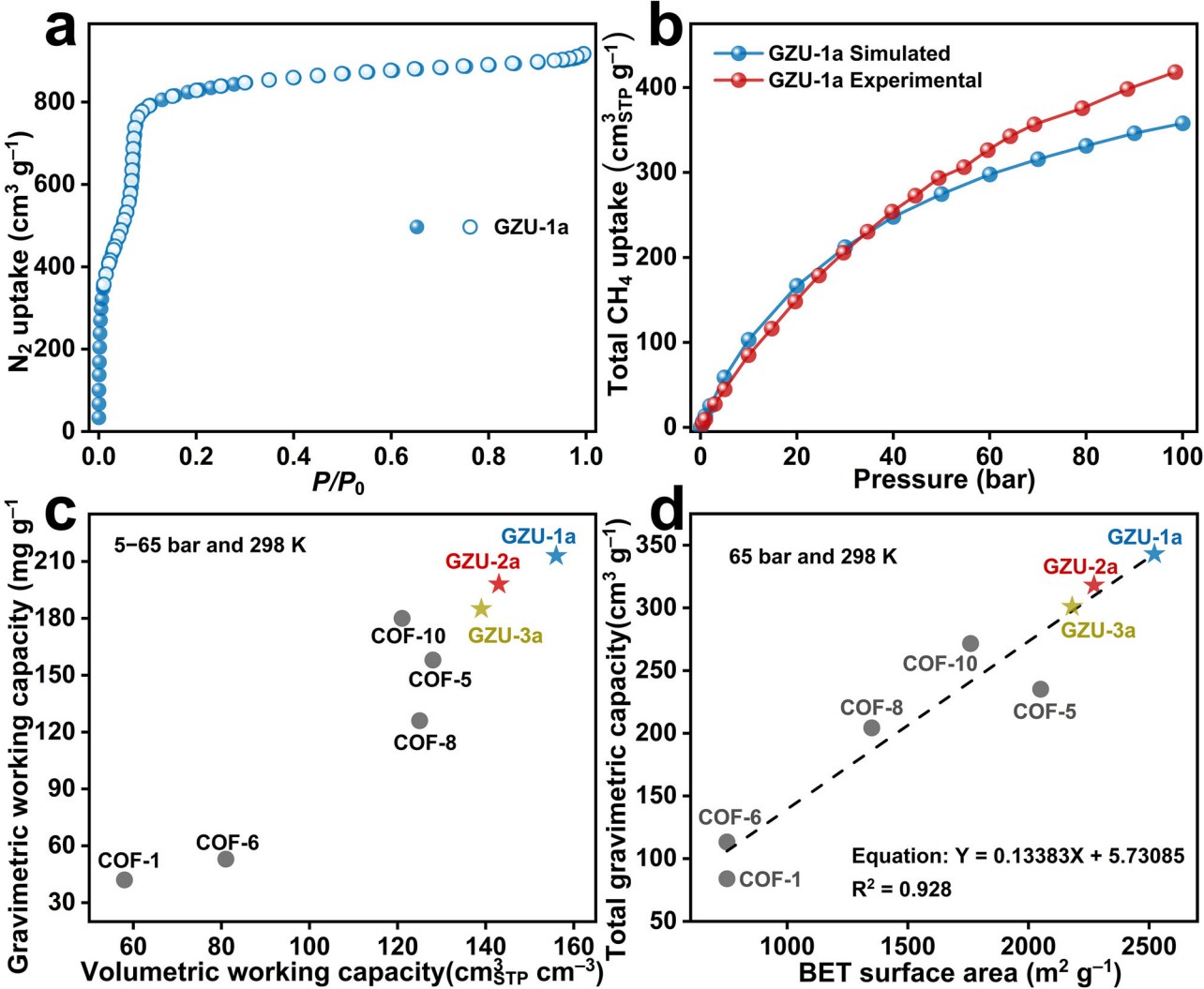

**Fig. 5 | Gas adsorption of single-crystal 2D COFs. a** The N₂ adsorption (solid) and desorption (hollow) isotherms of GZU-1a at 77 K. **b** The high-pressure total methane uptake of GZU-1a. **c** The methane gravimetric and volumetric working capacities of GZU-1a, GZU-2a, GZU-3a in comparison to other 2D COFs reported to date. **d** The relationship between the gravimetric methane storage capacities of the 2D COFs (grey dots) and the gravimetric BET surface area.

the calculated values from the simulated structures of 1.34, 1.40, of 1.41 cm³ g⁻¹ (Fig. 5a and Supplementary Figs. 40 and 41). In addition, the pore-size distribution curves of these three single-crystal COFs have been studied by adsorption isotherms employing the density functional theory method, displaying pores centered at 17.3 Å for GZU-1a, 17.0 Å for GZU-2a, and 16.1 Å for GZU-3a, Supplementary Figs. 46–48, aligning with the pore sizes obtained from the single-crystal structures (17.1 Å for GZU-1a, 16.9 Å for GZU-2a, and 16.2 Å for GZU-3a), Supplementary Figs. 49–51. These activated COFs afford moderate CO₂ absorption of 20, 25, and 26 cm³ g⁻¹ at room temperature and 1 bar for GZU-1a, GZU-2a, and GZU-3a due to the weak interactions between CO₂ molecules and COFs (Supplementary Figs. 52–54). These results further prove their permanent porosity.

**High-pressure methane storage of activated COFs**

Considering the high permanent porosity and large pore volume of these activated COFs, high-pressure methane adsorption experiments were performed at 273 and 298 K (Fig. 5b and Supplementary Figs. 55, 56). The total gravimetric methane uptake of the activated GZU-1a amounts to 299 mg g⁻¹ (418 cm³ (STP) g⁻¹) at 298 K and 100 bar. While GZU-2a and GZU-3a with different packing mode exhibit total gravimetric methane uptake of 289 mg g⁻¹ (405 cm³ (STP) g⁻¹) and

264 mg g⁻¹ (370 cm³ (STP) g⁻¹), respectively, under the same conditions. The gravimetrically higher methane uptake for GZU-1a than GZU-2a and GZU-3a is associated with its highest gravimetric BET surface area and pore volume among the three COFs. Force field grand Canonical Monte Carlo (GCMC) simulations confirm the attractive methane sorption performance of these three COFs as highlighted in Fig. 5b and Supplementary Figs. 57–59. These simulations reveal that the high methane uptake is associated with the high accessible surface exhibited by the single-crystal COFs, with the periodic and multi-directional layer slipping introducing indentations between the COF layers that favor methane adsorption (Supplementary Fig. 60). Compared with the traditional AA and AB stacking models of 2D COFs, this special layer slipping mode of these three single-crystal COFs can increase the number and cooperative strength of C–H···π interactions between adjacent layers, which help maintain the framework rigidity during activation and high-pressure CH₄ adsorption.

On the basis of the crystal densities of 0.523, 0.518, and 0.537 cm g⁻¹ for GZU-1a, GZU-2a, and GZU-3a without guest molecules, the corresponding total volumetric methane uptakes are calculated as 218, 210, and 198 cm³ (STP) cm⁻³ at 298 K and 100 bar (Supplementary Tables 1–3). In particular for GZU-1a, its total volumetric methane uptake is 240 cm³ (STP) cm⁻³ at 273 K and 100 bar. After the high-

pressure $CH_4$ sorption experiment, three COFs display similar PXRD patterns to the as-prepared species (Supplementary Figs. 61–63), further disclosing the rigid structures of these three COFs with multiple weak C−H···π interactions.

According to the methane isotherms obtained at 273 and 298 K, the adsorption isosteric heat of GZU-1a is calculated as 13.2 kJ mol$^{-1}$ (Supplementary Figs. 55, 56, 64), in agreement with the simulated value of 11.4 kJ mol$^{-1}$ and indicating a weak binding interaction between methane and the COF framework[4,6]. This is also true for GZU-2a and GZU-3a, with the adsorption isosteric heats of 12.3 and 11.8 kJ mol$^{-1}$ (Supplementary Fig. 64), also showing good consistency with the simulated values of 12.6 and 12.7 kJ mol$^{-1}$, respectively. This leads to the reduced $CH_4$ capacities at 5 bar and 298 K for GZU-1a (32 mg g$^{-1}$), GZU-2a (29 mg g$^{-1}$), GZU-3a (30 mg g$^{-1}$) (Fig. 5b and Supplementary Fig. 65). This, in combination with the high gravimetric methane uptake of GZU-1a (245 mg g$^{-1}$), GZU-2a (227 mg g$^{-1}$), and GZU-3a (215 mg g$^{-1}$) at 65 bar and 298 K, results in the usable methane gravimetric working capacities of 213, 198, and 185 mg g$^{-1}$, respectively. It is worth noting that the total gravimetric methane capacity of GZU-1 at 65 bar and 298 K is lower than that for many excellent MOFs absorbents such as UTSA-76a (263 mg g$^{-1}$)[7], NOTT-101a (247 mg g$^{-1}$)[50], MAF-38 (247 mg g$^{-1}$)[11], and NJU-Bai 42 (254 mg g$^{-1}$)[11] (Supplementary Figs. 65–67)[9–11]. However, its lower adsorption capacity at 5 bar than the above-mentioned MOFs remarkably induces high gravimetric working capacity (Fig. 5c and Supplementary Fig. 67). Noticeably, introduction of substituents onto the COFs frameworks seems to be able to reduce the $CH_4$ affinity at low pressure to enhance the gravimetric working capacity. Indeed, it must be pointed out that thus far, an increasing number of COFs have been synthesized and structurally characterized. However, high-pressure methane storage studies over COFs have been rarely explored due to the lack of single-crystal structural information to evaluate their performances. On the basis of the present result with the help of those reported previously, herein we established the correlation between total gravimetric methane uptake ($C_{total}$, cm$^3$ (STP) g$^{-1}$) at 65 bar and 298 K and experimental gravimetric BET area ($S$, m$^2$ g$^{-1}$), showing an overall linear relationship with an empirical formula of $C_{total} = 0.13383 \times S + 5.73085$ ($R = 0.93$), similar to that for MOFs[6] (Fig. 5d).

According to the crystal densities of COFs, the methane volumetric working capacities (5 to 65 bar) of GZU-1a, GZU-2a, and GZU-3a at 298 K are calculated as 156, 143, and 139 cm$^3$ (STP) cm$^{-3}$, respectively, (Fig. 5c). It is noteworthy that both methane volumetric and gravimetric working capacities (5 to 65 bar and 298 K) of these three COFs are significantly superior to those for the thus far reported 2D COFs (Fig. 5c). Moreover, both the methane volumetric and gravimetric working capacities (5 to 65 bar and 298 K) of GZU-1a are higher than those for GZU-2a and GZU-3a due to its larger experimentally revealed BET area and pore volume than the latter two species. In addition, the methane volumetric working capacities (5 to 65 bar and 298 K) of GZU-1a (156 cm$^3$ (STP) cm$^{-3}$) are even comparable with those for 3D COFs like COF-103 (165 cm$^3$ (STP) cm$^{-3}$)[10] and TAM-TFS-COF (157 cm$^3$ (STP) cm$^{-3}$)[12] (Supplementary Fig. 68). The above phenomena clearly indicate the bright perspective of 2D COFs in the methane storage since the further enlargement of pore size of **hcb-b** topological COFs enables the enhanced BET area and pore volume.

In summary, a substituent strategy has been proposed to design and prepare three single-crystal 2D **hcb-b** COFs isomers with different imine orientation and stacking modes. The atom-resolution crystal structures of these three COFs have been determined using 3D ED. Among the three compounds, GZU-1 and GZU-3 show an unprecedented ABCDEF 6-fold alternating stacking mode (AA$l_6$). Moreover, GZU-2 and GZU-3, with different imine orientations from GZU-1, exhibit the first example of stacking isomers with atom-resolution structures in single-crystal 2D COFs, and GZU-2 depicts the first unidirectional ABCD 4-fold tilted stacking structure (AA$t_4$). Moreover, fine

adjustment of interlayered distance of COFs at atomic precision results in the highest total volumetric methane uptake of 240 cm$^3$ (STP) cm$^{-3}$ at 273 K and 100 bar for the activated sample. In particular, GZU-1a possesses the highest methane gravimetric working capacity (5 to 65 bar at 298 K) of 213 mg g$^{-1}$, outperforming many state-of-the-art porous materials. The exceptional methane storage capacity of 2D single-crystal COFs demonstrates that the present structure-dependent property is originated from their precisely engineered interlayer packing. These findings will stimulate an improved understanding and leveraging of 2D covalent porous materials in multifunctional applications.

## Methods
### Synthesis of GZU-1
A 20 mL vial was loaded with TTFB (112.5 mg, 0.26 mmol), aniline (4.0 mL), and dry mesitylene (10.0 mL). The mixture was subjected to ultrasonication until complete dissolution of the solid components. This was followed by the addition of glacial acetic acid (2.0 mL). Subsequently, a solution of TOAB (114.8 mg, 0.26 mmol) in dry 1,4-dioxane (5.0 mL) was introduced into this reaction system. The resulting mixture was left undisturbed at 25 °C for 15 days, affording a large quantity of GZU-1 single crystals with sizes up to 10 μm. The pristine GZU-1 sample was successively solvent-exchanged with 1,4-dioxane, acetone, and $n$-hexane for 24 h, with the solvent renewed every 8 h. After this process, the material was collected by filtration and desolvated under dynamic vacuum at 100 °C for 12 h, yielding the activated material (147.5 mg, 71%).

### Synthesis of GZU-2
A 20 mL vial was loaded with TOFB (124.9 mg, 0.26 mmol), aniline (1.5 mL), and dry TCM (5.0 mL). The mixture was subjected to ultrasonication until complete dissolution of the solid components, then glacial acetic acid (2.0 mL) was added. Subsequently, a solution of TTAB (102.3 mg, 0.26 mmol) in dry TCM (5.0 mL) was immediately introduced into this reaction system. The resulting mixture was left undisturbed at 25 °C for 10 days, affording a large quantity of GZU-2 single crystals with sizes up to 5 μm. The pristine GZU-2 sample was successively solvent-exchanged with TCM and $n$-hexane for 24 h, with the solvent renewed every 8 h. The solvent-exchanged sample was then collected by filtration and desolvated under dynamic vacuum at 100 °C for 12 h, yielding the activated material (110.1 mg, 52%).

### Synthesis of GZU-3
A 20 mL vial was loaded with TOFB (124.9 mg, 0.26 mmol), aniline (4.0 mL), and dry mesitylene (10.0 mL). The mixture was subjected to ultrasonication until complete dissolution of the solid components. This was followed by the addition of glacial acetic acid (2.0 mL). Subsequently, a solution of TTAB (102.3 mg, 0.26 mmol) in dry 1,4-dioxane (5.0 mL) was immediately introduced into this reaction system. The resulting mixture was left undisturbed at 25 °C for 15 days, yielding a large quantity of GZU-3 single crystals with sizes up to 5 μm. The pristine GZU-3 sample was successively solvent-exchanged with 1,4-dioxane, acetone, and $n$-hexane for 24 h, with the solvent renewed every 8 h. The solvent-exchanged sample was then collected by filtration and desolvated under dynamic vacuum at 100 °C for 12 h, affording the activated material (132.9 mg, 63%).

### Density functional theory calculations
All Density functional theory (DFT) calculations were carried out using the Perdew−Burke−Ernzerhof (PBE) exchange−correlation functional[51] with DFT-D3(BJ)[52,53] dispersion corrections using the Quickstep module implemented in the CP2K package (version 2024.3)[54]. GTH pseudopotentials[55,56] were employed in conjunction with triple-ζ Gaussian basis sets augmented with one set of polarization functions, together with an auxiliary plane-wave basis set using a cutoff

energy of 1200 Ry mapped onto a five-level multigrid. The orbital transformation (OT) method was applied for electronic structure optimization of all models[57]. To assess the relative stability of different stacking isomers of the GZU COFs, various stacking configurations were constructed using the pyCOFBuilder package[58], as illustrated in Supplementary Fig. 26, and systematically compared with the experimentally determined structures.

### Grand canonical Monte Carlo (GCMC) simulations

All methane adsorption isotherms were simulated using the RASPA2 package[59,60]. For each pressure point of the isotherm, 50000 Monte Carlo cycles were conducted, and the equilibrium adsorption uptake was determined by averaging over the equilibrated portion of the trajectory, as identified using the pyMSER package[61]. Methane molecules were described using Lennard–Jones parameters from the TraPPE-UA force field[62], while COF framework atoms were modeled with the DREIDING force field[63]. Cross-interaction parameters were derived using the Lorentz–Berthelot mixing rules, and all Lennard–Jones interactions were truncated and shifted to zero at a cutoff distance of 12.8 Å. All simulations were performed using supercells large enough to ensure that each lattice vector exceeds the short-range interaction cutoff, thereby eliminating interactions between periodic images. Adsorption enthalpy was evaluated based on the fluctuations of energy and particle number in the grand canonical ensemble.

### Textural properties assessment

Pore size distribution, pore volume, and gravimetric surface area were calculated using the Zeo++ software (v0.3)[64,65]. A probe molecule with a radius of 1.86 Å was employed, and all calculations were performed in the High Accuracy mode with 100000 sampling cycles.

## Data availability

Crystallographic data reported in this paper are provided in the supplementary materials and archived at the Cambridge Crystallographic Data Centre under reference numbers 2455725 (GZU-1), 2415331 (GZU-2), and 2415329 (GZU-3). Copies of the data can be obtained free of charge from https://www.ccdc.cam.ac.uk/structures/. All other data are available in the main text or the supplementary materials. In addition, the data that support the findings of this study and the raw data for all the figures have been uploaded to Figshare at https://doi.org/10.6084/m9.figshare.30576989. Source data are provided with this paper. All data are available from the corresponding author upon request. Source data are provided with this paper.

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

## Acknowledgements

This work was financially supported by the Natural Science Foundation of China (Nos. 22235001 [J.J.], 22175020 [J.J.], 22131005 [H.W.], 22505047 [B.Y.], W2431013 [B.C.], and 22261132512 [H.W.]), Supported by the Postdoctoral Fellowship Program and China Postdoctoral Science Foundation (BX20250107 [B.Y.]), Xiaomi Young Scholar Program, the Fundamental Research Funds for the Central Universities (FRF-KST-25-006 [H.W.]), the Guizhou Provincial Key Laboratory Platform Project (ZSYS [2025] 008 [X.X.]), Talent Program of Guizhou University (LJ-2024-03) [J.J.], Guizhou University, and University of Science and Technology Beijing. We also thank the staff from the BL17B1 beamline of the National Facility for Protein Science in Shanghai (NFPS) at Shanghai Synchrotron Radiation Facility, for assistance during data collection. In addition, we acknowledge ReadCrystal Technology Co. for their prominent works in MicroED analysis. We are also very grateful for the help and discussion from Prof. Rui-Biao Lin at the Sun Yat-Sen University.

## Author contributions

B.Y., H.W., X.X., B.C., and J.J. conceived the subject and designed the experiments. B.Y. synthesized the materials and performed most of the characterization experiments. B.Y. measured the SEM of the COFs under

the supervision of Q.X and X.D. B.Y., H.W., H.L., S.Y., and Y.J. solved and analyzed the crystal structures. F.L.O., W.L, J.Z, and G.M. carried out the theoretical calculation of 2D COFs. B.Y., F.L.O., H.W., X.X., G.M., B.C., and J.J. interpreted the results and wrote the paper. All authors discussed the results and commented on the manuscript.

## Competing interests

The authors declare no competing interests.
