## [Transparent Peer Review file · Nature Communications]

Single-crystal 2D Covalent Organic Frameworks for High-capacity Methane Storage

Corresponding Author: Professor Jianzhuang Jiang

Version 0:

Reviewer comments:

Reviewer #1

(Remarks to the Author)

This manuscript describes synthesis of three single-crystal 2D COFs using a substituent-assisted strategy. Their single crystal structures are determined by 3D ED. Combining gas sorption, DFT, and GCMC, the authors reveal very high surface areas and methane uptakes/working capacities that are competitive with leading porous adsorbents. The topic and results should be interesting for the readers of Nat. Commun. However, there are several scientific flaws need further study and clarification. In addition, there are some overstatements which require modification. Therefore, I suggest a revision based on the current state of the manuscript. The detailed comments and questions are listed as following.

1. In the abstract, the authors mentioned "..., significant 2D COF structure-originated superiority in methane storage." The authors need to elaborate the mechanism about why the 6-fold stacking has the advantage over e.g., AB stacking, ABC stacking, etc. In addition, the structure-originated "superiority" is overstated, as there are reported 3D COFs and MOFs shows better methane storage performance (Science, 2024, 386, 693; J. Am. Chem. Soc. 2014, 17, 6207, etc.).
2. The authors mentioned ""The N₂ sorption isotherms of these three COFs possess the Type IV form, ...". According to IUPAC, a Type IV isotherm is characteristic of mesopores with capillary condensation and often hysteresis. None of those presents in the isotherm of the COFs. The authors need to revise this part and the following claims.
3. The authors report a total volumetric CH₄ uptake of 240 cm³ (STP) cm⁻³ at 273 K and 100 bar for GZU-1a, and claim "very close to the DOE target". However, the DOE target of 263 cm³ (STP) cm⁻³ is explicitly defined for 65 bar and ~298 K (room temperature). Therefore, the statement from the authors is misleading and need to be revised.
4. The authors concluded "..., GZU-1 exhibit the first example of stacking isomers in single-crystal 2D COFs ...". However, COF isomers with different stacking modes have been reported (ACS Appl. Mater. Interfaces 2021, 13, 29471; J. Mater. Chem. A, 2025, 13, 27661). The authors need to reconsider their claim about the first.
5. In the Supplementary Information, the authors calculated "formation energies" per COF sheet for GZU and other COFs. The authors further claim that the more negative values "highlight the high thermodynamic stability of the GZU series compared to other structurally related frameworks, facilitating the growth of high-quality single crystals." In solvothermal COF synthesis, kinetics, solubility, reversibility of bond formation, etc., are all critical. Claiming that more exergonic formation energies "facilitate the growth of high-quality single crystals" is oversimplified.
6. While the authors have done it partly, they need to address all A and B alerts in the checkcif files.

Reviewer #2

(Remarks to the Author)

Baoqiu Yu et al. report the synthesis of three new imine-linked COFs based on methyl- and methoxy-substituted triphenylbenzene monomers. The COFs were obtained as small single crystals via a modulated synthesis using aniline. Single crystal structures were obtained via 3D electron diffraction (3DED). They reveal rather complex stacking patterns of slightly shifted COF layers. The COFs are highly porous with BET surface areas well above 2000 m² g⁻¹ and methane storage capacities of up to 213 mg g⁻¹ at 298 K and 5 – 65 bar.

2D COF single crystals have remained scarce due to their challenging synthesis. Structure analysis of honeycomb lattice 2D COFs with their complex layer stacking is an important finding. There has been indirect evidence that these materials are generally more complex than the simple AA stacking pictures that are commonly used to describe most 2D COFs, but the direct evidence here from single crystal data will certainly increase the understanding of these COF geometries. However, I was wondering if the authors could quantify the density of stacking faults – I assume they should be quite common for ABCDEF layer sequences?

The materials are highly porous with very well-defined N₂ sorption isotherms and high BET surface areas. However, I could not find the pore size distributions. These should be added.

Regarding the BET surface areas, the ranges where the BET model was applied might be incorrect. BET plots are up to 0.05 which is in the region of the mesopore filling step. However, the BET model applies only to the multilayer formation without capillary condensation effects. Typically one will find two linear regions in the BET plots of mesoporous COFs and it would be the one at lower relative pressure that should be used.

GZU 2 seems to change slightly upon vacuum activation. But the single crystal structure is also recorded under high vacuum, and its simulation seems closer to the PXRD of the activated COF. I think the patterns used for the PXRD refinements in Figure 2 should be the ones of the activated (solvent-free) COFs.

Minor points:

Line 80 should read methoxy (not methoxyl) substituents

Suppl figure 41 – please plot separately or shifted to be able to see individual isotherms

In summary, this is an important and convincing study. A couple of technical aspects should be updated/corrected as pointed out above. After these changes I recommend the publication in Nature Communications.

Reviewer #3

(Remarks to the Author)

I enjoyed reading this interesting article titled "Single-crystal 2D Covalent Organic Frameworks for High-capacity Methane Storage" by Yu et al. The authors propose and demonstrate a substituent strategy to prepare three single-crystal 2D hcb-b COFs isomers with different imine orientation and stacking modes. The high degree of crystallinity evidenced by the e-diffraction is noteworthy and as explained by the authors, this is quite rare in 2D COFs. The structural modeling is well done and brings important design aspects in layer stacking. The importance of the functional side groups in fine-tuning the stacking to achieve high methane storage is valuable. The authors have identified through computational modeling the superior formation energy arising from the efficient molecular interactions and also comparison with other related COF structure has been used to elucidate this. I think the figures from the SI need to be moved into the main text for easy readability and this part is important to the work. Currently, the DFT studies identify the higher crystallinity of these phases via formation energy which is attributable more to the periodic lattice, however, an alternate modeling at molecular level could help identify the impact of the side groups such as -OCH₃ in influencing the packing and thereby the order. This can be included to garner further insights. Overall, this is a neat well-investigated work on a high-quality COF material contributing further to this interesting class of 2D COFs. I recommend its publication in Nature Communications.

Version 1:

Reviewer comments:

Reviewer #1

(Remarks to the Author)

The authors have addressed all my comments. I am happy with the improvements being made and recommend the manuscript for publication.

Reviewer #2

(Remarks to the Author)

Baoqiu Yu et al. report the synthesis of three new imine-linked COFs based on methyl- and methoxy-substituted triphenylbenzene monomers. The COFs were obtained as small single crystals via a modulated synthesis using aniline. Single crystal structures were obtained via 3D electron diffraction (3DED). They reveal rather complex stacking patterns of slightly shifted COF layers. The COFs are highly porous with BET surface areas well above 2000 m² g⁻¹ and methane storage capacities of up to 213 mg g⁻¹ at 298 K and 5 – 65 bar. BET analysis and pore size distributions have been updated/added.

The authors have fully addressed my comments. I suggest the publication without further changes.

Reviewer #3

(Remarks to the Author)

The revisions are satisfactory and has certainly improved the manuscript's thoroughness. Authors have responded to all the technical queries with rational explanations. I recommend its publication in this revised form.

Response to reviewers' reports:

Reviewer 1:

Comment: *This manuscript describes synthesis of three single-crystal 2D COFs using a substituent-assisted strategy. Their single crystal structures are determined by 3D ED. Combining gas sorption, DFT, and GCMC, the authors reveal very high surface areas and methane uptakes/working capacities that are competitive with leading porous adsorbents. The topic and results should be interesting for the readers of Nat. Commun. However, there are several scientific flaws need further study and clarification. In addition, there are some overstatements which require modification. Therefore, I suggest a revision based on the current state of the manuscript. The detailed comments and questions are listed as following.*

Response: Thanks a lot for the time and effort of this reviewer in reviewing our paper. We are also very grateful for his/her encouragements and professional comments towards further improving the quality of this paper.

(1) In the abstract, the authors mentioned "..., significant 2D COF structure-originated superiority in methane storage." The authors need to elaborate the mechanism about why the 6-fold stacking has the advantage over e.g., AB stacking, ABC stacking, etc. In addition, the structure-originated "superiority" is overstated, as there are reported 3D COFs and MOFs shows better methane storage performance (Science, 2024, 386, 693; J. Am. Chem. Soc. 2014, 17, 6207, etc.).

Response: Thanks a lot for this professional comment. According to Density Functional Theory (DFT) calculations, the ABCDEF 6-fold stacking structure of GZU-1 is the most energetically stable one, being approximately 62, 73, and 122-141 kJ mol⁻¹ per sheet more stable than the slip AAl, the eclipsed AA, and the staggered AB patterns, respectively, Supplementary Fig. 26 and Supplementary Table 4. In addition, the 6-fold stacking structure of GZU-1 introduces a larger degree of interlayer slip and generates periodic interlayer indentations that expose additional adsorption-accessible surface. Moreover, this stacking mode can increase the number and cooperative strength of C–H··· π interactions between adjacent layers, which help maintain the framework rigidity during activation and high-pressure CH₄ adsorption.

As pointed out by this reviewer, 3D COFs and MOFs show remarkable advantages for high-pressure methane storage and have been widely studied. It is worth noting that different from their MOFs counterparts which are dominated by the three-dimensional (3D) extended frameworks, most of the reported COFs are of two-dimensional (2D) layered structures. This seems to retard the exploration over their methane storage application due to the still lack of report on high methane storage performance for 2D MOFs associated with their low porosities and instability. Fortunately, the late

construction of highly mesoporous 2D COFs with pore volume as high as $2.7 \text{ cm}^3 \text{ g}^{-1}$ motivates us to explore 2D COFs for their methane storage functionality (*Adv. Mater.* **2020**, *32*, 1905776). In addition, following the instruction of this reviewer, we have removed the expression “structure-originated superiority,” which might be interpreted as overstated. The corresponding result has been corrected in the revised version of manuscript.

(2) *The authors mentioned ““The N₂ sorption isotherms of these three COFs possess the Type IV form, ...”. According to IUPAC, a Type IV isotherm is characteristic of mesopores with capillary condensation and often hysteresis. None of those presents in the isotherm of the COFs. The authors need to revise this part and the following claims.*

Response: Thank this reviewer very much for this important clarification. We fully agree with this reviewer in the point that the N₂ sorption isotherms of these three COFs at 77 K do not exhibit the capillary condensation or hysteresis feature characteristic of IUPAC Type IV isotherm. As a consequence, we have revised the manuscript to remove the Type IV classification. Instead we re-describe in this way “The N₂ adsorption isotherms of these three COFs at 77 K display a pronounced two-step uptake behavior. A sharp increase in adsorption isotherms is observed at very low relative pressures ($P/P_0 < 0.01$), followed by a more gradual uptake in the range of $P/P_0 \approx 0.01$ – 0.05 . Upon further increasing the relative pressure, the other sharp increase of adsorption appears in the range of $P/P_0 \approx 0.05$ – 0.1 , achieving a saturated adsorption.” This sorption behavior is similar to the mesoporous COFs with pore size close to 2 nm. The above behavior allows these three COFs to be classified as quasi-mesoporous materials. All subsequent statements that rely on the previous Type IV assignment in the present revised version of manuscript have also been corrected to ensure consistency with the IUPAC classification. These revisions do not affect the conclusion regarding the permanent porosity of the materials. In addition, the pore-size distribution curves of these three single-crystal COFs have been studied by adsorption isotherms employing the density functional theory method, displaying pores centered at 17.3 Å for GZU-1a, 17.0 Å for GZU-2a, and 16.1 Å for GZU-3a, Supplementary Figs. 44–46, aligning with the pore sizes obtained from the single-crystal structures (17.1 Å for GZU-1a, 16.9 Å for GZU-2a, and 16.2 Å for GZU-3a), Supplementary Figs. 47–49. The above results further confirm that these three COFs are quasi-mesoporous materials. Corresponding result has been added into the revised version of manuscript and Supporting Information.

(3) *The authors report a total volumetric CH₄ uptake of $240 \text{ cm}^3 \text{ (STP)} \text{ cm}^{-3}$ at 273 K and 100 bar for GZU-1a, and claim “very close to the DOE target”. However, the DOE target of $263 \text{ cm}^3 \text{ (STP)} \text{ cm}^{-3}$ is explicitly defined for 65 bar and ~298 K (room temperature). Therefore, the statement from the authors is misleading and need to be revised.*

Response: We are indeed very sorry for having made such kind of mistakes. The incorrect comparisons have been removed in the revised manuscript. In addition, the whole manuscript has been carefully checked to avoid such kind of errors.

(4) The authors concluded “ ..., GZU-1 exhibit the first example of stacking isomers in single-crystal 2D COFs ...”. However, COF isomers with different stacking modes have been reported (ACS Appl. Mater. Interfaces 2021, 13, 29471; J. Mater. Chem. A, 2025, 13, 27661). The authors need to reconsider their claim about the first.

Response: Good point. Just as pointed out by this reviewer, COF isomers with different stacking modes have been reported. The precise control over the interlayer stacking mode for 2D COFs is a challenging pursuit due to the complicated and uncontrollable interactions of 2D sheets. To the best of our knowledge, about 10 cases of work focusing on the stacking mode control of 2D COFs have been reported thus far (refs: *Nat. Commun.* **2024**, *15*, 194; *J. Am. Chem. Soc.* **2023**, *145*, 21798-21806; *J. Am. Chem. Soc.* **2022**, *144*, 20363-20371; *ACS Appl. Mater. Interfaces* **2021**, *13*, 29471; *Adv. Mater.* **2025**, *37*, 2500468; *Angew. Chem. Int. Ed.* **2025**, *64*, e202512603; and *J. Mater. Chem. A*, **2025**, *13*, 27661, etc.). Notably, these aforementioned COFs structures were identified through PXRD studies with the help of theoretical simulations, thus bringing somehow structural uncertainties due to their polycrystalline nature. As a result, constructing single-crystal 2D COFs with atom-resolution single-crystal structures and modulating their interlayer stacking patterns are important in this field. These 2D COF isomers (GZU-1, GZU-2, and GZU-3) feature atom-resolution structures determined by 3D electron diffraction, a precise level of structural elucidation not achieved in previous studies. To avoid any potential ambiguity, we have added further clarification in the revised version of manuscript.

(5) In the Supplementary Information, the authors calculated “formation energies” per COF sheet for GZU and other COFs. The authors further claim that the more negative values “highlight the high thermodynamic stability of the GZU series compared to other structurally related frameworks, facilitating the growth of high-quality single crystals.” In solvothermal COF synthesis, kinetics, solubility, reversibility of bond formation, etc., are all critical. Claiming that more exergonic formation energies “facilitate the growth of high-quality single crystals” is oversimplified.

Response: Thanks a lot for this important and insightful comment. We fully agree with this reviewer in the point that COF crystallization is governed by a complex interplay of factors including reaction kinetics, monomer solubility, bond reversibility, and nucleation–growth dynamics, which cannot be determined solely by thermodynamic formation energies. Our previous wording suggesting that more exergonic formation energies directly “facilitate the growth of high-quality single crystals” is indeed an oversimplification. In the revised version of Supplementary

Information, we have removed this causal implication and now restrict the discussion of formation energies to their role as indicators of relative thermodynamic stability among different stacking arrangements and frameworks, which might help rationalize the observed preference for certain packing modes.

(6) *While the authors have done it partly, they need to address all A and B alerts in the checkcif files.*

Response: Thanks a lot for pointing out this matter. According to this instruction, we have carefully revisited all A and B alerts in the CheckCIF reports and provided detailed explanations in the revised crystallographic data. Many of these alerts originate from the intrinsic limitations of 3D electron diffraction (3D ED) and the weakly scattering, beam-sensitive nature of 2D single-crystal COFs. Unlike X-ray diffraction, 3D ED data collection is typically constrained by limited tilting ranges, reduced high-angle completeness, and unavoidable dynamical scattering effects, all of which can trigger alerts related to completeness, data redundancy, or refinement restraints.

We have also refined the structures and reprocessed the data multiple times, making every reasonable effort to reduce or eliminate these alerts (such as the anisotropic refinement). However, despite extensive attempts and considerable effort, some A and B alerts could not be completely removed due to the less data collected by 3D ED and poor single-crystal quality of 2D COFs. For example, (1) RINTA01_ALERT_3_A, the value of Rint is greater than 0.25; (2) PLAT020_ALERT_3_A, the value of Rint is greater than 0.12. As can be easily expected, for single-crystal COFs characterized by 3D ED, the presence of such alerts is widely recognized as common and often unavoidable (refs: *Nat. Chem.* **2025** *17*, 571–581; *Nat. Chem.* **2025**, *17*, 226–232; *J. Am. Chem. Soc.* **2023**, *145*, 13537–13541; *Chem* **2024**, *10*, 2170–2179; *J. Am. Chem. Soc.* **2023**, *145*, 25332–25340; and .etc), reflecting the experimental constraints rather than structural inaccuracies. To ensure transparency, we have now provided explicit justifications for each alert in the revised CIF files. Importantly, the remaining alerts do not compromise the validity of the structural models, which are strongly supported by consistent PXRD patterns, pore characteristics, and DFT calculations.

Reviewer 2:

Comment: *Baoqiu Yu et al. report the synthesis of three new imine-linked COFs based on methyl- and methoxy-substituted triphenylbenzene monomers. The COFs were obtained as small single crystals via a modulated synthesis using aniline. Single crystal structures were obtained via 3D electron diffraction (3D ED). They reveal rather complex stacking patterns of slightly shifted COF layers. The COFs are highly porous with BET surface areas well above 2000 m² g⁻¹ and methane storage capacities of up to 213 mg g⁻¹ at 298 K and 5–65 bar.*

2D COF single crystals have remained scarce due to their challenging synthesis. Structure analysis of honeycomb lattice 2D COFs with their complex layer stacking is an important finding. There has been indirect evidence that these materials are generally more complex than the simple AA stacking pictures that are commonly used to describe most 2D COFs, but the direct evidence here from single crystal data will certainly increase the understanding of these COF geometries.

Response: Many thanks for the time and effort of this reviewer paid in reviewing this paper. As can be found in the revised version of manuscript and Supporting Information, all the comments have been thoroughly addressed accordingly.

(1) However, I was wondering if the authors could quantify the density of stacking faults – I assume they should be quite common for ABCDEF layer sequences?

Response: Quite interesting question, thanks a lot. Actually it is hard to quantify the density of stacking faults for the 2D COFs reported thus far due to the lack of their either single crystal X ray diffraction analysis- or 3D ED analysis-determined structures. On the other hand, without either single crystal X ray diffraction analysis- or 3D ED analysis-determined structures, at this stage we cannot say that those 2D COFs with structures determined on the basis of powder X ray diffraction analysis with the help of theoretical modeling are truly correct or accurate. At the end of this paragraph, we would like to emphasize that for most previously reported 2D COFs, their structural models were derived from powder X-ray diffraction combined with theoretical simulations, which inherently provide averaged structural information (ref: *Chem. Sci.*, **2020**, *11*, 12647–12654).

(2) The materials are highly porous with very well-defined N₂ sorption isotherms and high BET surface areas. However, I could not find the pore size distributions. These should be added.

Response: Thanks a lot for this very nice advice! According to the instruction, the pore-size distribution curves of these three single-crystal COFs have been studied by adsorption isotherms employing the density functional theory method, displaying pores centered at 17.3 Å for GZU-1a, 17.0 Å for GZU-2a, and 16.1 Å for GZU-3a, Supplementary Figs. 44–46, aligning with the pore sizes obtained from the single-crystal structures (17.1 Å for GZU-1a, 16.9 Å for GZU-2a, and 16.2 Å for GZU-3a), Supplementary Figs. 47–49. It is worth noting again that The N₂ adsorption isotherms of these three COFs at 77 K display a pronounced two-step uptake behavior. A sharp increase in adsorption isotherms is observed at very low relative pressures ($P/P_0 < 0.01$), followed by a more gradual uptake in the range of $P/P_0 \approx 0.01–0.05$. Upon further increasing the relative pressure, the other sharp increase of adsorption appears in the range of $P/P_0 \approx 0.05–0.1$, achieving a saturated adsorption, Fig. 5a and Supplementary Figs. 38 and 39. This sorption behavior is similar to the mesoporous

COFs with pore size close to 2 nm. Corresponding result has been added into the revised version of manuscript and Supporting Information.

(3) Regarding the BET surface areas, the ranges where the BET model was applied might be incorrect. BET plots are up to 0.05 which is in the region of the mesopore filling step. However, the BET model applies only to the multilayer formation without capillary condensation effects. Typically one will find two linear regions in the BET plots of mesoporous COFs and it would be the one at lower relative pressure that should be used.

Response: Thanks again for this good suggestion towards further improving the quality of this work. On the basis of this instruction, the BET plots and reassessed fitting ranges of these three COFs have been re-examined to ensure the utilization of a single-layer N₂ filling range. To avoid fitting errors in BET calculations, the Rouquerol criteria have also been applied, revealing the range of single-layer N₂ filling is up to 0.046 for GZU-1a, 0.048 for GZU-2a, and 0.048 for GZU-3a, respectively. As shown is Supplementary Figs. 40–42 (Figs. R1–R3 also given below), the linear regions at lower relative pressure are used to calculate the BET surface areas of activated GZU-1, GZU-2, and GZU-3, which are 2100, 1965, and 1897 m² g⁻¹, respectively. Corresponding figures and data have been updated in the revised version of manuscript and Supporting Information. These corrections do not alter the overall trends or the structure–property relationships discussed in the work.

Figure R1. The BET surface area calculation for GZU-1a based on the N₂ adsorption isotherm at 77 K. (a) Rouquerol plot (red dot was selected for BET surface area calculation). (b) BET plot taken from $P/P_0 = 0.0015 - 0.046$. [C : the BET constant; V_m : the monolayer capacity]

The following values obtained for the BET Surface Area, Slope, Y-Intercept, C , V_m , R^2 , $1/(\sqrt{C} + 1)$, and P/P_0 at V_m Using the Fittings.

BET surface area	$2099.8700 \pm 39.3132 \text{ m}^2 \text{ g}^{-1}$
Slope	$0.002066 \pm 0.000039 \text{ g cm}^{-3}$
Y-intercept	$0.000007 \pm 0.000001 \text{ g cm}^{-3}$

C	317.785372
V_m	482.4428 cm ³ g ⁻¹
R^2	0.998
$1/(\sqrt{C} + 1)$	0.0531
P/P_0 at V_m	0.0432

Figure R2. The BET surface area calculation for GZU-1a based on the N₂ adsorption isotherm at 77 K. (a) Rouquerol plot (red dot was selected for BET surface area calculation). (b) BET plot taken from $P/P_0 = 0.0002 - 0.048$.

The following values obtained for the BET Surface Area, Slope, Y-Intercept, C , V_m , R^2 , $1/(\sqrt{C} + 1)$, and P/P_0 at V_m Using the Fittings. [C : the BET constant; V_m : the monolayer capacity]

BET surface area	1965.0422 ± 36.1635 m ² g ⁻¹
Slope	0.002209 ± 0.000041 g cm ⁻³
Y-intercept	0.000006 ± 0.000001 g cm ⁻³
C	400.628167
V_m	482.4428 cm ³ g ⁻¹
R^2	0.998
$1/(\sqrt{C} + 1)$	0.0476
P/P_0 at V_m	0.0411

Figure R3. The BET surface area calculation for GZU-1a based on the N_2 adsorption isotherm at 77 K. (a) Rouquerol plot (red dot was selected for BET surface area calculation). (b) BET plot taken from $P/P_0 = 0.0003 - 0.048$.

The following values obtained for the BET Surface Area, Slope, Y-Intercept, C , V_m , R^2 , $1/(\sqrt{C} + 1)$, and P/P_0 at V_m Using the Fittings. [C : the BET constant; V_m : the monolayer capacity]

BET surface area	$1897.1534 \pm 30.8696 \text{ m}^2 \text{ g}^{-1}$
Slope	$0.002289 \pm 0.000037 \text{ g cm}^{-3}$
Y-intercept	$0.000005 \pm 0.000001 \text{ g cm}^{-3}$
C	450.471415
V_m	$435.8689 \text{ cm}^3 \text{ g}^{-1}$
R^2	0.998
$1/(\sqrt{C} + 1)$	0.0450
P/P_0 at V_m	0.0397

(4) GZU 2 seems to change slightly upon vacuum activation. But the single crystal structure is also recorded under high vacuum, and its simulation seems closer to the PXRD of the activated COF. I think the patterns used for the PXRD refinements in Figure 2 should be the ones of the activated (solvent-free) COFs.

Response: Thanks a lot for this careful and technically insightful observation and good suggestion. As pointed out by the reviewer, the single-crystal structure of these three COFs was collected under high-vacuum conditions, and the simulated PXRD patterns derived from the solvent-free structures show better agreement with the PXRD of the activated COFs. In the revised manuscript, we have clarified that the PXRD refinements in Figure 2 are based on the solvent-free/activated structures. These revisions improve consistency between the structural models and the experimental PXRD data.

(5) Line 80 should read methoxy (not methoxyl) substituents

Response: We are indeed very sorry for having made such kind of mistakes, which have been corrected in the revised version of manuscript. In addition, the whole manuscript has been carefully checked to avoid such kind of errors.

(6) Suppl figure 41 – please plot separately or shifted to be able to see individual isotherms

Response: Thanks again for this good suggestion. As can be seen in Fig. 5a and Supplementary Figs. 38 and 39, the N₂ sorption isotherms at 77 K of these three activated single-crystal COFs have been presented separately for clarity in the revised version of Supporting Information.

(7) In summary, this is an important and convincing study. A couple of technical aspects should be updated/corrected as pointed out above. After these changes I recommend the publication in Nature Communications.

Response: Thanks a lot for the time and effort of this reviewer in reviewing our paper. We are also very grateful for his/her encouragements and professional comments towards further improving the quality of this paper.

Reviewer 3:

Comment: I enjoyed reading this interesting article titled "Single-crystal 2D Covalent Organic Frameworks for High-capacity Methane Storage" by Yu et al. The authors propose and demonstrate a substituent strategy to prepare three single-crystal 2D hcb-b COFs isomers with different imine orientation and stacking modes. The high degree of crystallinity evidenced by the *e*-diffraction is noteworthy and as explained by the authors, this is quite rare in 2D COFs. The structural modeling is well done and brings important design aspects in layer stacking. The importance of the functional side groups in fine-tuning the stacking to achieve high methane storage is valuable. The authors have identified through computational modeling the superior formation energy arising from the efficient molecular interactions and also comparison with other related COF structure has been used to elucidate this. I think the figures from the SI need to be moved into the main text for easy readability and this part is important to the work. Currently, the DFT studies identify the higher crystallinity of these phases via formation energy which is attributable more to the periodic lattice. However, an alternate modeling at molecular level could help identify the impact of the side groups such as -OCH₃ in influencing the packing and thereby the order. This can be included to garner further insights. Overall, this is a neat well-investigated work on a high-quality COF material contributing further to

this interesting class of 2D COFs. I recommend its publication in Nature Communications.

Response: Many thanks for the time paid in reviewing this paper as well as the professional comments and encouragements of this reviewer. According to the instruction, the figure comparing the formation energy between the GZU COFs with similar structures reported previously has been added into the revised version of manuscript from the Supporting Information.

Figure R4. Non-covalent interactions (NCI) surface plots of the between GZU-1 COF layers. (a) The main interactions between the layers arises from the $\text{CH}_3 \cdots \pi$ from the OMe groups and benzene from adjacent layers, represented in green, and the T-shaped benzene-benzene interactions, represented in yellow. Other small lateral interactions between the adjacent layers are represented in blue. Perpendicular (b) and in-plane (c) views with all the interactions between the layers. gray: carbon, white: hydrogen, red: oxygen, and blue: nitrogen.

In addition, based on the suggestion of this reviewer, we perform additional simulations to elucidate the impact of the OMe side groups on the intermolecular interactions between the COF layers, and thereby its impact on the overall crystallinity. Specifically, we performed non-covalent interaction (NCI) plot analysis based on the QTAIM theory (ref: *Comput. Phys. Commun.*, **2014**, 185, 1007–1018.) using the electronic density for both the crystal structure and molecular models.

This new result helps confirm that the high crystallinity of the GZU COFs series is driven by specific interlayer interactions that function as "conformational lockers".

NCI analysis reveals that the layers are stabilized by $\text{CH}_3 \cdots \pi$ interactions between methoxy groups and aromatic rings, alongside T-shaped benzene-benzene interactions, Supplementary Fig. 37 (Fig. R4 also given above). These forces, which are stronger than standard $\pi \cdots \pi$ stacking, dictate the precise orientation of the building blocks during the crystallization process. This structural motif of GZU-1 provides a representative explanation for the superior structural order found throughout this family of materials.

We are very much grateful for the reviewer's overall positive evaluation and recommendation for publication. The insightful comments have helped us improve the clarity, presentation, and mechanistic discussion of the manuscript, and we believe the revised version more clearly conveys the significance of this work to the broader COF field.

Response to reviewers' reports:

Reviewer 1:

Comment: The authors have addressed all my comments. I am happy with the improvements being made and recommend the manuscript for publication.

Response: Many thanks for the time and efforts paid by this reviewer in reviewing this paper.

Reviewer 2:

Comment: Baoqiu Yu et al. report the synthesis of three new imine-linked COFs based on methyl- and methoxy-substituted triphenylbenzene monomers. The COFs were obtained as small single crystals via a modulated synthesis using aniline. Single crystal structures were obtained via 3D electron diffraction (3DED). They reveal rather complex stacking patterns of slightly shifted COF layers. The COFs are highly porous with BET surface areas well above 2000 m² g⁻¹ and methane storage capacities of up to 213 mg g⁻¹ at 298 K and 5–65 bar.

BET analysis and pore size distributions have been updated/added.

The authors have fully addressed my comments. I suggest the publication without further changes.

Response: Thanks a lot for the time and efforts of this reviewer in reviewing our paper again.

Reviewer 3:

Comment: The revisions are satisfactory and has certainly improved the manuscript's thoroughness. Authors have responded to all the technical queries with rational explanations. I recommend its publication in this revised form.

Response: Many thanks for the time paid in reviewing this paper as well as the encouragements of this reviewer.